# Temporal Temperature Distribution in Shallow Sediments of a Large Shallow Lake and Estimated Hyporheic Flux Using VFLUX 2 Model

**Yong Li** [1,2,*], **Na Li** [1,2,*], **Jiacheng Feng** [2], **Jianing Qian** [2] and **Yajie Shan** [2]

1 Key Laboratory of Integrated Regulation and Resources Development on Shallow Lakes, Ministry of Education, Hohai University, Nanjing 210098, China
2 College of Environment, Hohai University, Nanjing 210098, China; student_fengran@163.com (J.F.); qianjiang97@163.com (J.Q.); 13815530040@163.com (Y.S.)
* Correspondence: liyonghh@hhu.edu.cn (Y.L.); hhulina@163.com (N.L.)

**Abstract:** Identifying and quantifying exchange flux across sediment-water interface is crucial when considering water and nutrient contributions to a eutrophic lake. In this study, observed temporal temperature distributions in shallow sediment of Lake Taihu (Eastern China) based on three-depth sensors at 14 sites throughout 2016 were used to assess temporal water exchange patterns. Results show that temporal temperature in shallow sediments differed with sampling sites and depths and the temperature amplitudes also clearly shrunk as the offshore distance increasing. Exchange fluxes estimated using the VFLUX 2 model based on temperature amplitude show that alternating-direction temporal flow exists in the eastern zone of Lake Taihu with averages of −13.0, −0.6, and 3.4 mm day$^{-1}$ (negative represents discharging into the lake) at three nearshore sites (0.5, 2.0, and 6.0 km away from the shoreline, respectively). Whereas downwelling flow occurred throughout almost the entire year with averages of 37.7, 23.5, and 6.6 mm day$^{-1}$ at the three southern nearshore sites, respectively. However, upwelling flow occurred throughout almost the entire year and varied widely in the western zone with averages of −74.8, 45.9, and −27.0 mm day$^{-1}$ and in the northern zone with averages of −76.2, −55.3, and −51.1 mm day$^{-1}$. The estimated fluxes in the central zone were relatively low and varied slightly during the entire year (−15.1 to 22.5 mm day$^{-1}$ with an average of −0.7 mm day$^{-1}$). Compared with the sub sensor pair (at 5 and 10 cm), the estimated hyporheic fluxes based on the top sensor pair (at 0 and 5 cm) varied within wider ranges and exhibited relatively larger values. Effects of upwelling flow at the western and northern zones need to be paid attention to on nearshore water quality particularly during winter and spring seasons. Estimated flow patterns at the four zones summarily reflect the seasonal water interaction near the sediment surface of Lake Taihu and are beneficial to improve its comprehensive management. Thermal dispersivity usually used for estimating the thermal diffusivity is more sensitive for upward hyporheic flux estimating even if with a low flux. Temperature amplitude ratio method can be used to estimate the exchange flux and suitable for low flux conditions (either upwelling or downwelling). A better evaluation of the exchange flux near inclined nearshore zones might need an optimized installation of temperature sensors along with the potential flow path and/or a vertical two-dimensional model in the future.

**Keywords:** shallow sediment; temporal temperature distribution; flow pattern; exchange flux; VFLUX 2 model; Lake Taihu

## 1. Introduction

The hyporheic zone is an active hydraulical transition zone connecting surface water and groundwater, with enhanced gradients in hydraulic pressure, water flux, temperature, redox condition, and nutrient concentration [1–3]. These special distributions of environmental factors and their interactions are mostly resulted from complex flow patterns in the hyporheic zone. In general, the hyporheic flow patterns are particularly crucial for

some inner and then interacting processes such as nutrient exchange and energy transfer. Characterizing and quantifying the flow patterns is of great importance in efforts to better understand the nutrient cycling near sediment-water interface (SWI) and its influence on surface water. Considerable efforts, including pressure head difference [4–6] and conservative tracer methods [7,8], have been applied in recent years to ascertain the hyporheic flow patterns, especially in the hyporheic zone of rivers/streams.

Heat as a natural tracer was widely applied to discriminate daily/seasonally flow direction and even to quantify flux dynamics [4,9]. Temperature (heat) is easy to measure and model, making it a practical tool for studying hyporheic flow patterns [10,11]. Generally, vertical flow in the hyporheic zone is the most dominant controlling factor on vertical temperature profile. Static, upwelling, or downwelling flow patterns, as well as the seepage velocity, generally result in diverse temperature profiles near the SWI [12,13]. As a result, temporal temperature–depth profiles in the hyporheic zone are widely used to inversely distinguish the hyporheic flow direction and to quantify exchange flux [14–16].

In recent years, increased studies have built upon the correlation between water flow and heat transfer in the hyporheix zone to deduce the exchange flux using mathematical models. Anibas et al. [17] used inverse modeling (STRIVE model) of the one-dimensional heat transport equation to estimate vertical advective fluxes based on spatially and seasonally distributed temperature profiles in the hyporheic zone of the Aa River (Belgium). Angermann et al. [18] used controlled laboratory experiments to explore the accuracy of analytical solutions in a one-dimensional heat transport model for capturing temporal variability of flux through porous media from propagation of a periodic temperature signal to depth. Irvine et al. [19] experimentally determined exchange flux and thermal diffusivity from temperature time series using VFLUX 2 model based on phase, amplitude, and combined methods.

However, limited documentation has been published on lakes particularly a large shallow lake in a plain region, because of the relatively low water flux through the hyporheic zone into lakes compared to rivers/streams [20–22]. Generally, the hyporheic zone near lakeshore is active in water exchange and nutrient transport, having an important effect on nearshore water quality [23–25]. Furthermore, the temperature distribution near the SWI has an important effect on the redistribution and exchange of nutrients [20,26,27]. Many previous studies [28–30] have proved that the sediment of some lakes served as a source or sink of nutrients into the lake that even seasonally switched locations, mostly depending on exchange flow directions near the SWI. It is necessary to know whether the temporal temperature profiles are further effective to estimate exchange flow direction and flux magnitude and more reasonable to explain the source/sink switch in a lake.

In order to verify the exchange flow direction and quantify the fluxes near the SWI of shallow Lake Taihu, temporal temperatures at shallow sediments in different zones were observed during 2016 based on three temperature sensors at 0, 5, and 10 cm depths. Exchange fluxes at different sites were estimated and compared using a widely used model (VFLUX 2) with a method of temperature amplitude ratios. The rationality of estimated exchange fluxes in the eastern, southern, western, northern, and central zones were also analyzed. Additionally, the disparity of estimated hyporheic fluxes between top and sub temperature sensor pairs were discussed as well as the sensitivity of thermal parameters on the estimation of exchange flux.

## 2. Materials and Methods

### 2.1. Study Site Description

Lake Taihu, the third-largest freshwater lake in China, is a shallow subtropical lake located in the Yangtze River Delta of Eastern China (Figure 1a). It covers a water area of 2338 km$^2$, with maximum and mean water depths of approximately 3.0 and 1.9 m, respectively [31]. As many as 117 main rivers and tributaries flow into Lake Taihu. Annual freshwater input to the lake reaches approximately $8.8 \times 10^9$ m$^3$, mostly through western and northern rivers. Lake water leaves mainly through southern and eastern rivers (e.g.,

Taipu River) [31,32]. The hydrodynamics of Lake Taihu are also dominated by frequent wind-induced currents which have notable effects on water and nutrient mixing [33,34].

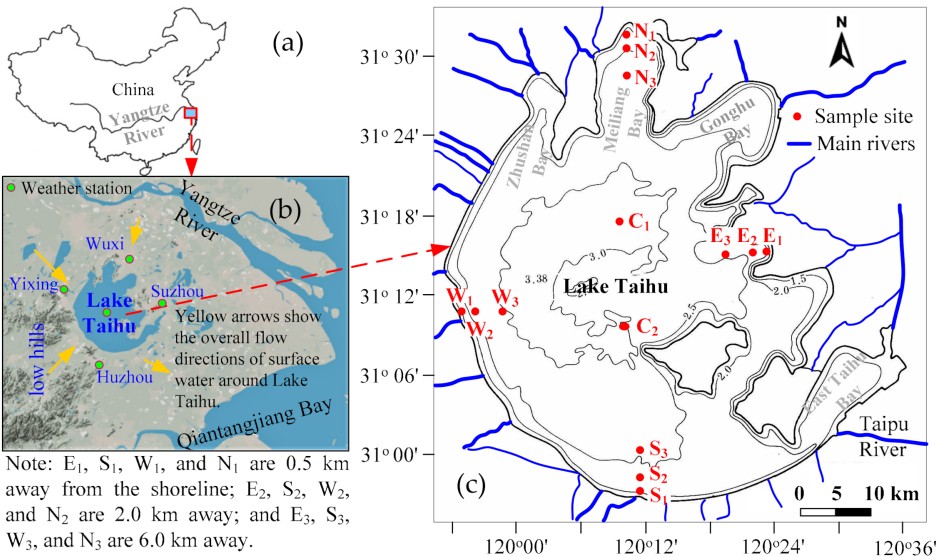

**Figure 1.** (**a**) Location of Lake Taihu in China; (**b**) schematic topography and overall surface water flow directions (yellow arrows) around the lake and weather stations; (**c**) main rivers (blue lines) around the lake and sampling sites (red dots) of the hyporheic zone.

The whole basin topography resembles a shallow dish with Lake Taihu located at the central (Figure 1b). Except the low mountain chains at west and southwest of the lake, most of the topography consists of plain and river-networks in the western and northern regions, plain and small lakes and river networks in the eastern region, and plains in the southern region. The hyporheic zone in the nearshore zone of Lake Taihu is mostly composed of alluvial sediments, while faint yellow clay sediments are present in the offshore zone of the lake [35]. Groundwater table surrounding the lake are mostly about 0.2–1.2 m below the land surface and vary seasonally with precipitation and irrigation events. The basin has a subtropical monsoon climate with average annual rainfall of 1,181 mm, 60% of which occurs from May through September. Annual mean water temperature in Lake Taihu is about 17.1 °C, whereas the monthly mean water temperature peaks at a maximum value in July (about 25 °C) and a minimum value in January (about 4 °C).

The lake is surrounded by developed cities and towns and intensively cultivated agricultural land. Wastes from surrounding industrial, agricultural and municipal sewage enter the lake, causing severe environmental problems. Water quality and the aquatic ecosystem of Lake Taihu have severely degraded in recent decades [23,36], and eutrophication has consequently become serious [31,37]. The nutrients just released from sediments still sufficiently deteriorate the water environment according to other previous studies [38,39]. Presently, although significant measures have been undertaken, the water quality in Lake Taihu has shown little improvement, and algal blooms still occur locally during summer.

*2.2. Measurements and Analysis*

Shallow sediments (about 0–12.5 cm) were sampled at 14 sites in Lake Taihu, including 12 near the shoreline at eastern, southern, western, and northern zones and 2 in the central (Figure 1c) according to some previous studies about the sediment feature distribution and potential exchange flow zone [23,29,31,40]. Three sites in each nearshore zone were set along a transect perpendicular to the shore. The first site (i.e., $E_1$, $S_1$, $W_1$, and $N_1$) in each nearshore zone was 0.5 km away from the shoreline, the second site (i.e., $E_2$, $S_2$, $W_2$, and $N_2$) was 2.0 km away, and the third (i.e., $E_3$, $S_3$, $W_3$, and $N_3$) was 6.0 km away. Three sediment samples were collected at each site within a 10 m range using Plexiglas tubes

(6.5 cm in diameter) that were then sliced into several sub-samples. The sediments between about 0–2.5 cm represent the surface sediment samples, and those at about 2.5–7.5 cm and 7.5–12.5 cm, respectively, represent the 5 and 10 cm depth sediment samples. All sediment samples were enclosed in black plastic bags, stored on ice, and transported to the laboratory within 2 h for further analysis.

Thermal properties of undisturbed sediment samples (0–5 cm and 5–10 cm, respectively), including thermal diffusivity ($\kappa_e$), volumetric heat capacity($C$), and thermal conductivity ($\lambda_0$) of the saturated sediment, were in situ analyzed using a portable thermal properties analyzer (KD2 Pro, Decagon Devices, Inc., Pullman, WA, USA) with an SH1 Probe. Sediment porosity ($n$) and bulk density (BD) were measured using a cutting ring by weighing 50 cm$^3$ fresh sediments after drying overnight at 105 °C [41]. The volumetric water content $\theta$ was quantified by weighing sediment specimens immediately after sampling and after they had been dried for 24 h at 105 °C. Disturbed sediment samples were used to analyze grain size distribution of sediments using a Laser Particle Analyzer (LS13320, Beckman Coulter Co., Brea, CA, USA).

Temporal temperatures were measured in the field during 2016 at sediment surface (in bottom water, about 5–10 cm above sediment surface, indicated as sediment surface) and depths of 5 and 10 cm of the 14 sampling sites (near the sediment sampling sites) using a self-made instrument with three temperature sensors (0.1 °C). The temperature data were recorded once every four hours and collected weekly after adjusting a starting time. Daily air temperature data were obtained from five weather stations (Wuxi, Yixing, Suzhou, Huzhou and central stations, Figure 1b).

### 2.3. VFLUX 2 Model for Estimating Hyporheic Flux

The VFLUX 2 model, a program written in the MATLAB computing language (distributed as a toolbox), was used to estimate the hyporheic fluxes near SWI at the 14 sampling sites according to the temporal temperature distributions. VFLUX 2 model calculates one-dimensional vertical fluid flow (seepage flux) through saturated porous media using heat transport equations developed by Hatch et al. [4] and Keery et al. [42]. It uses temperature time series data measured by multiple temperature sensors in a vertical profile in order to calculate flux at specific times and depths. VFLUX 2 processes the temperature time series by first resampling the data to a lower sampling rate, in order to reduce the filtering problems associated with oversampling. The program then filters each time series using Dynamic Harmonic Regression programs, isolating a fundamental temperature signal identified by its period of oscillation (typically a diurnal signal). Finally, the program calculates vertical flux between pairs of temperature sensors using the amplitude ratios of their filtered temperature signals, according to Hatch et al. [4] and Keery et al. [42]. Details about VFLUX 2 model could be found in other published references [19,43].

The one-dimensional vertical heat transport equation is

$$\frac{\partial T}{\partial t} = \kappa_e \frac{\partial^2 T}{\partial z^2} - q \frac{C_w}{C} \frac{\partial T}{\partial z} \tag{1}$$

where $T$ is temperature (°C), $t$ is time (s), $q$ is fluid flux (m s$^{-1}$, positive downward), $C_w$ is the volumetric heat capacity of water ($4.2 \times 10^6$ J m$^{-3}$ °C$^{-1}$), $C$ is volumetric heat capacity of saturated sediment (J m$^{-3}$ °C$^{-1}$), and $z$ is depth (m), and $\kappa_e$ is thermal diffusivity of the saturated sediment (m$^2$ s$^{-1}$), calculated by

$$\kappa_e = \frac{\lambda_0}{C} + \beta \left( \frac{C_w |q|}{C} \right) \tag{2}$$

where $\lambda_0$ is the thermal conductivity of the bulk saturated medium (W m$^{-1}$ °C$^{-1}$), $\beta$ is thermal dispersivity (m), and here were considered equal to 0.001 for all analyses in this study.



Amplitude variations provide more reliable estimates of relatively low seepage rates (e.g., $< \pm 1.0$ m d$^{-1}$) than do the variations in phase [4]. Hatch et al. [4] provided an analytical solution for the vertical water flux between two depth sensors as a function of amplitude ratios between the sensors' temperature signals:

$$q = \frac{C}{C_w}\left(\frac{2k_e}{\Delta z}\ln A_r + \sqrt{\frac{\alpha + v_t^2}{2}}\right) \tag{3}$$

$$\alpha = \sqrt{v_t^4 + \left(\frac{8\pi k_e}{P}\right)^2} \tag{4}$$

where $A_r$ is the ratio of amplitudes between the lower and upper sensors (= $A_{\text{lower}}/A_{\text{upper}}$), $\Delta z$ is the distance between the two depth sensors in the sediment (m), $v_t$ is the thermal front velocity (m s$^{-1}$), and $P$ is the period of the temperature signal (e.g., 86,400 s).

Measured temporal temperature data on the basis of top (at 0 and 5 cm depths, top-pair) and sub (at 5 and 10 cm depths, sub-pair) sensor pairs were used to distinguish hyporheic flow direction and estimate flux using VFLUX 2 based on the Hatch amplitude ratio ($A_r$) method [4]. A positive sign of the vertical groundwater velocity stands for downward flow (downwelling) from the surface into the hyporheic zone (e.g., groundwater recharge or losing lake sites) and a negative sign represents upward flow (upwelling flow) from the hyporheic zone into the surface water (e.g., groundwater discharge or gaining lake sites).

## 3. Results

### 3.1. Physical and Thermal Features of Shallow Sediment in Lake Taihu

Shallow sediment samples (about 0–12.5 cm) at the 14 sites of Lake Taihu have very similar grain size distributions (Figure 2). The sediments are mostly composed of a silt component, which ranges from 71.6% to 86.1% in volume (mean $\pm$ SD: 78.9% $\pm$ 3.1%), followed by a clay component of 6.4%–23.5% (12.6% $\pm$ 5.3%) and a sand component of 0–19.8% (8.5% $\pm$ 5.6%). Most sediment samples belong to a kind of silt loam soil, except that the surface sediments (0–2.5 cm) at sites $E_2$, $E_3$, $N_2$, $N_3$, and $C_2$, and those at a depth of 10 cm of $E_3$ belong to a kind of silt soil. In comparison, the sediments at eastern sites have a relative low clay component proportion (average 8.7%), and those at the central sites have a relative high clay proportion (19.4%). Bulk densities (BDs) of sediment samples range from 0.60 to 1.15 g cm$^{-3}$ (0.86 $\pm$ 0.13 g cm$^{-3}$) and increase with depth. On average, southern sites have relative high BD values (average 1.01 g cm$^{-3}$), whereas central sites have lower values (average 0.74 g cm$^{-3}$). However, no significant differences ($p < 0.05$) in the grain size and BD were found between the sampling zones.

Measured $C$, $\lambda_0$, and $\kappa_e$ of shallow sediments at the 14 sampling sites range from 3.25 to 3.92 MJ m$^{-3}$ K$^{-1}$ (3.52 $\pm$ 0.13 MJ m$^{-3}$ K$^{-1}$), from 1.02 to 1.30 W m$^{-1}$ K$^{-1}$ (1.15 $\pm$ 0.08 W m$^{-1}$ K$^{-1}$), and from 2.58 $\times$ 10$^{-7}$ to 3.83 $\times$ 10$^{-7}$ m$^2$ S$^{-1}$ (3.25 $\times$ 10$^{-7}$ $\pm$ 0.42 $\times$ 10$^{-7}$ m$^2$ S$^{-1}$), respectively (Figure 3). Relatively high mean $C$ values (Figure 3a) are exhibited in southern zones (mean 3.79 MJ m$^{-3}$ K$^{-1}$) and central zone (3.71 MJ m$^{-3}$ K$^{-1}$), and relative low mean values are exhibited in eastern zones (3.33 MJ m$^{-3}$ K$^{-1}$) and northern zone (3.37 MJ m$^{-3}$ K$^{-1}$). However, mean $\lambda_0$ and $\kappa_e$ values (Figure 3b,c) almost show an inverse spatial distribution with the $C$ values. The differences of thermal feature among sampling sites in a nearshore zone were not significant ($p < 0.05$).

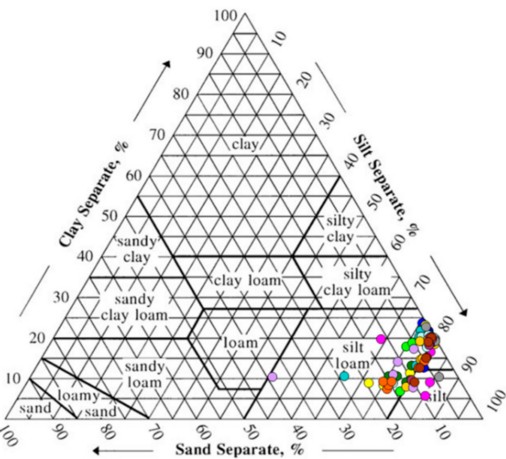

**Figure 2.** Grain size distributions of shallow sediments (0–12.5 cm) in Lake Taihu.

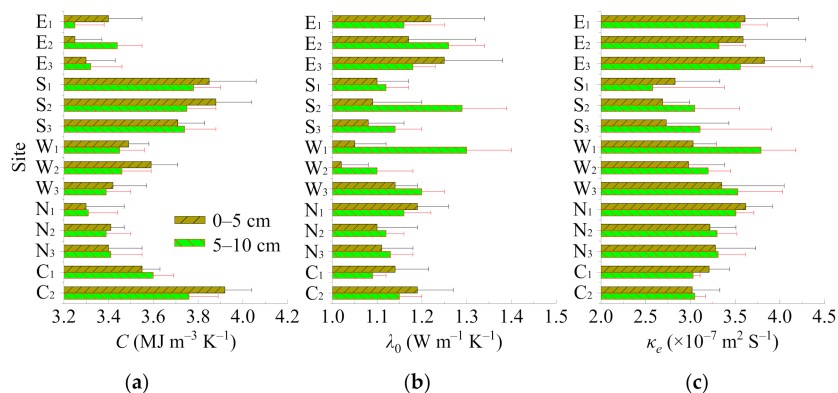

**Figure 3.** (**a**) In situ measured mean bulk heat capacity ($C$), (**b**) thermal conductivity ($\lambda_0$), and (**c**) thermal diffusivity ($\kappa_e$) of saturated sediments at the eastern, southern, western, northern, and central zones of Lake Taihu. Data are given as mean $\pm$ SD (measured times for each sample $n$ = 3–5), and only one-half of one standard deviation on one site shows here.

### 3.2. Temporal Temperature of Air and Shallow Sediment

Daily mean air temperature (Figure 4) during 2016 at eastern, southern, western, northern, and central sites varied between $-6.5$ °C and 35.5 °C (mean 17.9 °C), $-6.0$ °C and 34.0 °C (17.6 °C), $-8.0$ °C and 34.5 °C (17.4 °C), $-7.0$ °C and 34.5 °C (17.5 °C), and $-7.9$ °C and 33.5 °C (16.6 °C), respectively. Maximum and minimum air temperatures occurred in July (35.5 °C in the east) and January ($-8$ °C in the west), respectively.

Temporal temperatures of shallow sediment distributed within relatively wide variable ranges and varied seasonally (Figure 5). Measured daily mean temperatures at the SWI (about 5–10 cm above sediment surface) at eastern, southern, western, northern, and central sites had similar distributions, varying between 4.6 °C and 32.2 °C (mean 16.8 °C), 4.1 °C and 31.9 °C (mean 16.4 °C), 4.0 °C and 32.5 °C (mean 16.9 °C), 5.4 °C and 32.1 °C (mean 16.8 °C), and 3.7 °C and 29.6 °C (mean 16.0 °C), respectively. Maximum and minimum temperatures occurred on 30 July (32.5 °C at site $W_1$) and 7 February (3.7 °C at site $C_2$), respectively. It was apparent that the temperature at the interface varied with seasonal air temperature although with a slight hysteresis and decreased variations. Additionally, the temperature variation at each zone declined with offshore distance. Site $C_2$ in the central exhibited minimum temperature variations (3.7–29.6, 6.4–26.6, and 7.5–24.9 °C at depths of 0, 5, and 10 cm, respectively), following by site $C_1$ (5.4–29.2, 6.7–26.6, and 7.7–24.9 °C at depths of 0, 5, and 10 cm, respectively).

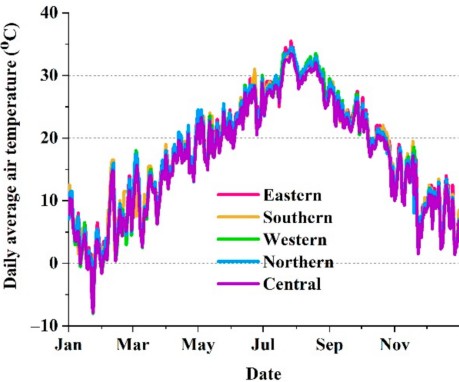

**Figure 4.** Daily average air temperatures during 2016 at the eastern (Suzhou), southern (Huzhou), western (Yixing), northern (Wuxi), and the central weather stations of Lake Taihu (see Figure 1b).

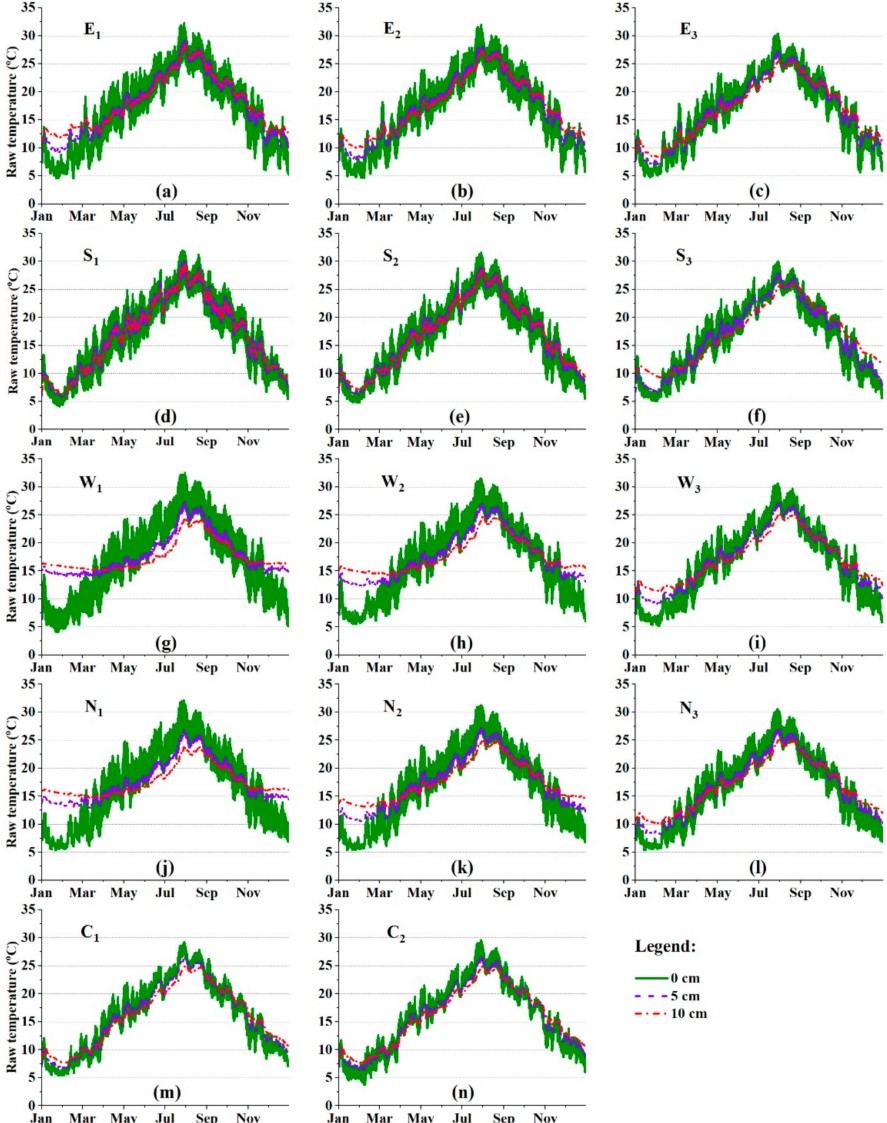

**Figure 5.** Temporal raw temperature at the sediment surface and depths of 5 and 10 cm of the eastern (**a–c**), southern (**d–f**), western (**g–i**), northern (**j–l**), and central sites (**m,n**) at Lake Taihu during 2016. Site locations see Figure 1c.

Temporal temperature in shallow sediments differed with depths and sites. Temperature distributions exhibited a large disparity between at the interface and at depths of 5 and 10 cm. As the depth increased, temperature variations obviously lagged and shrunk. Western, northern, and eastern sites (Figure 5g–l) exhibited greater difference of temperature distribution between depths in particular at those nearshore sites (e.g., at sites $W_1$ and $N_1$). Furthermore, the seasonal temperature variations at both 5 and 10 depths at western and northern sites were also apparently narrowed compared with those at southern and central sites. At southern sites, the temperature variations at the three depths (0, 5, and 10 cm) were almost synchronous, in particular at site $S_1$, and the relatively small disparities were mostly exhibited during the winter-spring period (Figure 5d–f). At the two central sites ($C_1$ and $C_2$, Figure 5m,n), temperature at the three depths had similar distributions and exhibited relatively narrower variations between the three depths compared with those at western and northern sites. However, three depths at almost all sites had similar temperature distributions but slightly lower values during April and October (Figure 5), during this period the decline of temperature at a deeper depth was apparent at western and northern sites.

### 3.3. Temperature Amplitude in Shallow Sediments

Overall, the temperature amplitude at the SWI of all sampling sites had a similar seasonal distribution, averagely ranging from 0.3 to 4.3 °C (mean 2.1 °C) at eastern zone, from 0.4 to 4.0 °C (2.2 °C) at southern zone, from 0.3 to 4.2 °C (2.2 °C) at western zone, from 0.2 to 4.3 °C (2.1 °C) at northern zone, and from 0.2 to 2.6 °C (1.2 °C) at central zone, respectively (Figure 6). However, smaller average temperature amplitudes at a depth of 5 cm were common throughout 2016 (2.8 ± 0.3 °C) particularly at the central sites (Figure 6m,n). Furthermore, the temperature amplitudes at a depth of 10 cm approached almost zero particularly at the western and northern sites (Figure 6g–l). Overall, relatively strong temperature oscillations transmitted to depths of 5 cm and even 10 cm at southern sites particularly at site $S_1$ (Figure 6d), whereas that transmitted less depth at western and northern sites.

Additionally, the temperature amplitudes also clearly shrunk as the offshore distance increasing. For example, the mean temperature amplitudes declined from 2.5 °C at site $E_1$ to 2.2 °C at $E_2$ and to 1.7 °C at $E_3$, and from 2.8 °C at site $W_1$ to 2.1 °C at $W_2$ and to 1.6 °C at $W_3$. Two central sites ($C_1$ and $C_2$) had smaller average temperature amplitudes compared to those nearshore sites. Compared to during the winter and summer seasons, the temperature amplitudes were relative higher during spring and autumn seasons.

### 3.4. Estimated Hyporheic Flow Flux Based On Temperature Amplitude Using VFLUX 2 Model

Based on the top-pair (0 and 5 cm depths), the estimated hyporheic fluxes ranged from −48.7 to 32.4 mm day$^{-1}$ (mean ± SD of −12.8 ± 15.8 mm day$^{-1}$) at site $E_1$ and from −24.4 and 25.2 mm day$^{-1}$ (−0.6 ± 9.1 mm day$^{-1}$) at site $E_2$, showing that the upwelling flux peaked in the winter–spring season but the downwelling flux peaked in Autumn (Figure 7a,b and Figure 8a). At site $E_3$, the fluxes varied between −23.7 and 37.6 mm day$^{-1}$ (3.5 ± 6.8 mm day$^{-1}$). Upwelling flow (negative) occurred during most of the year particularly at site $E_1$, whereas downwelling flow (positive) occurred mostly during the summer-autumn period (Figure 7a–c). At the three southern sites, the estimated fluxes based on the top-pair clearly showed that the hyporheic flow was mostly directed downwards and peaked in summer-autumn period (Figure 7d–f). Comparably, site $S_1$ (Figure 8b) had a maximum mean flux (37.7 ± 24.3 mm day$^{-1}$) and a wider variation range (−6.2 to 86.4 mm day$^{-1}$). Similar to eastern sites, the estimated hyporheic fluxes declined gradually from site $S_1$ to site $S_2$ (23.5 ± 10.6 mm day$^{-1}$, −3.1 to 51.0 mm day$^{-1}$) and site $S_3$ (6.7 ± 6.2 mm day$^{-1}$, −15.7 to 35.9 mm day$^{-1}$).

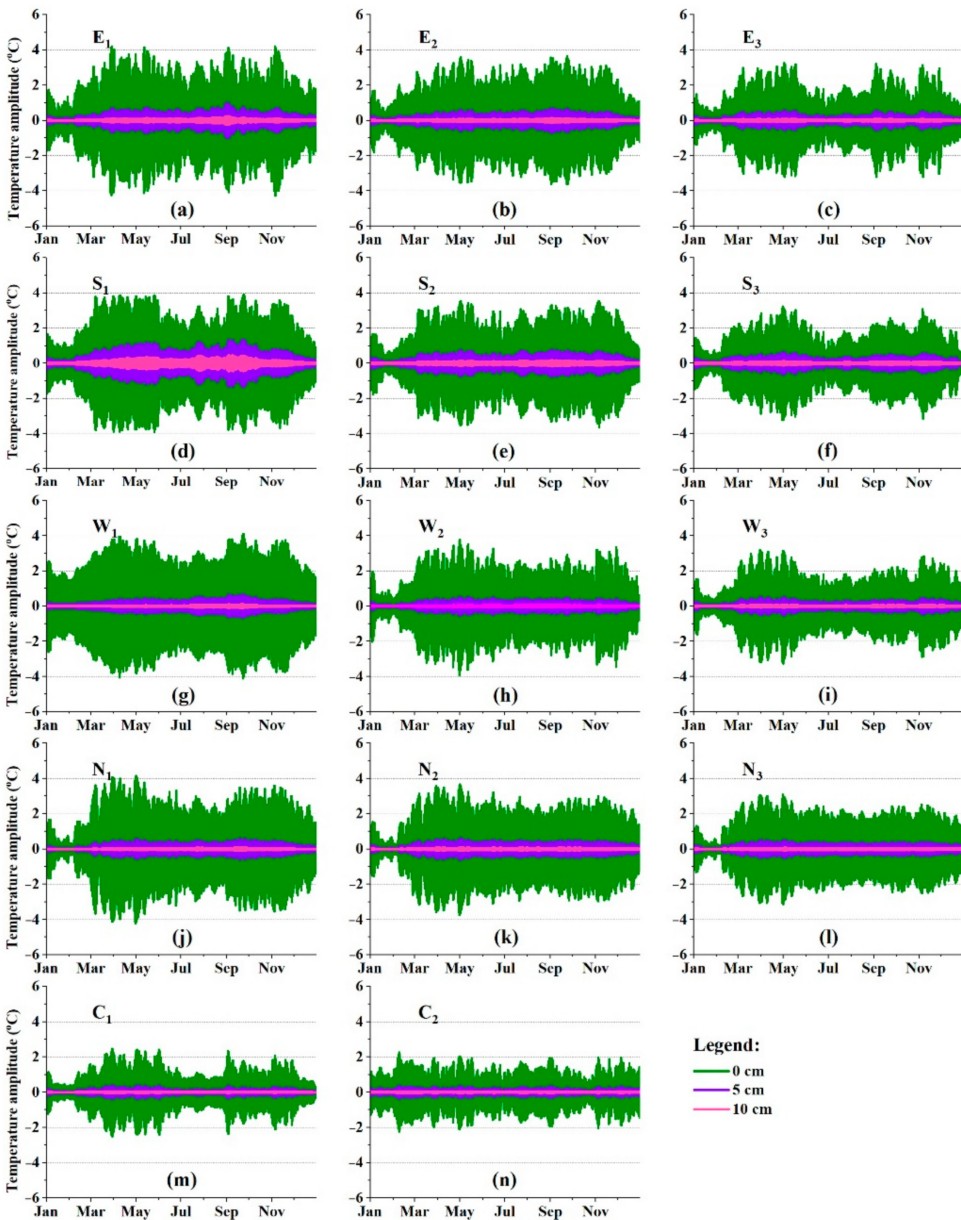

**Figure 6.** Temporal temperature amplitudes at sediment surface and depths of 5 and 10 cm of the eastern (**a–c**), southern (**d–f**), western (**g–i**), northern (**j–l**), and central sites (**m,n**) at Lake Taihu during 2016. Site locations see Figure 1c.

At western and northern sites, estimated hyporheic fluxes based on the top-pair had similar seasonal distribution and flow direction (upwards mostly) and also declined obviously with offshore distance (Figures 7g–l and 8c,d). The estimated fluxes ranged between −136.5 and −12.3 mm day$^{-1}$ (−74.5 ± 34.4 mm day$^{-1}$) at site $W_1$, between −94.7 and −4.1 mm day$^{-1}$ (−45.8 ± 19.2 mm day$^{-1}$) at site $W_2$, and between −61.6 and 18.0 mm day$^{-1}$ (−26.8 ± 10.8 mm day$^{-1}$) at site $W_3$ (Figures 7c and 8c). Those at sites $N_1$, $N_2$, and $N_3$ ranged from −131.2 to −29.3 mm day$^{-1}$ (−76.3 ± 23.9 mm day$^{-1}$), from −89.8 to −20.8 mm day$^{-1}$ (−55.1 ± 12.1 mm day$^{-1}$), and from −77.5 to −14.8 mm day$^{-1}$ (−51.0 ± 7.7 mm day$^{-1}$), respectively (Figures 7d and 8d). At the two central sites, estimated hyporheic fluxes based on the top-pair varied slightly around zero during the whole of 2016, ranging between −13.2 and 17.4 mm day$^{-1}$ (−0.6 ± 4.0 mm day$^{-1}$) at $C_1$ and between −13.6 and 17.4 mm day$^{-1}$ (−1.1 ± 4.0 mm day$^{-1}$) at $C_2$ (Figure 7m,n and Figure 8e).

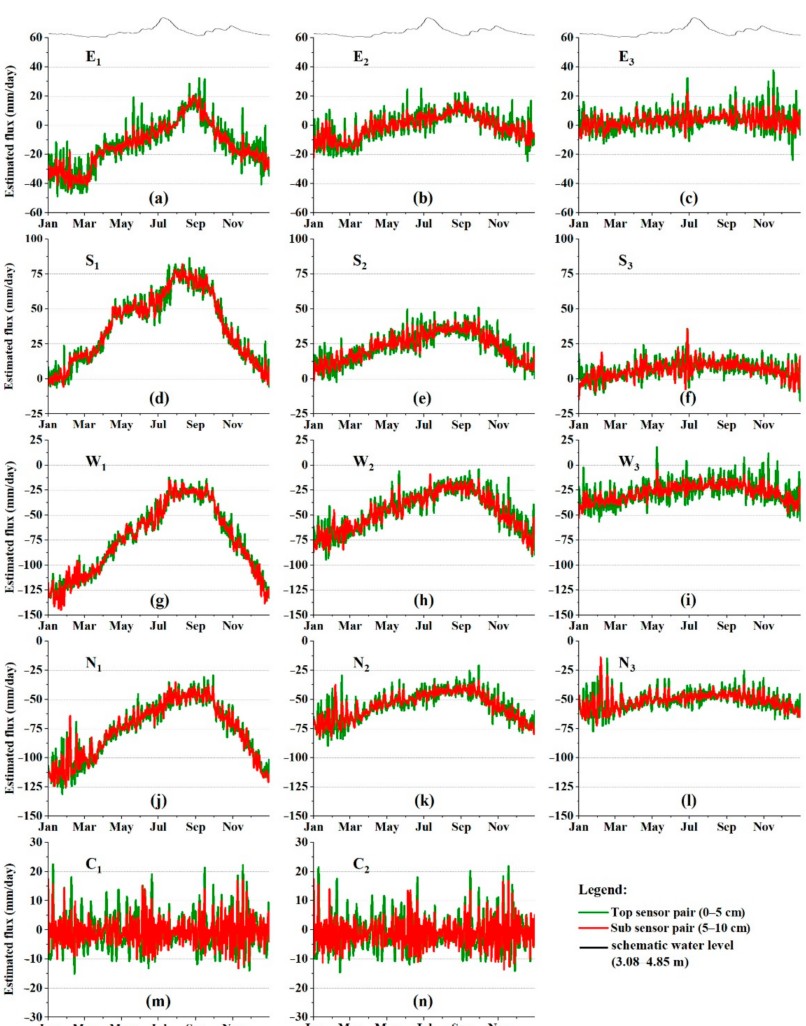

**Figure 7.** Estimated hyporheic flux over time at 14 sampling sites during 2016 using the VFLUX 2 model based on top (depths of 0 and 5 cm, green lines) and sub-top (depths of 5 and 10 cm, red lines) sensor pairs. Positive flux values represent downward flow and negative values represent upward flow. Site locations see Figure 1c. Schematic mean water level of Lake Taihu (3.08–4.85 m) was shown at the top of this figure (gray solid line).

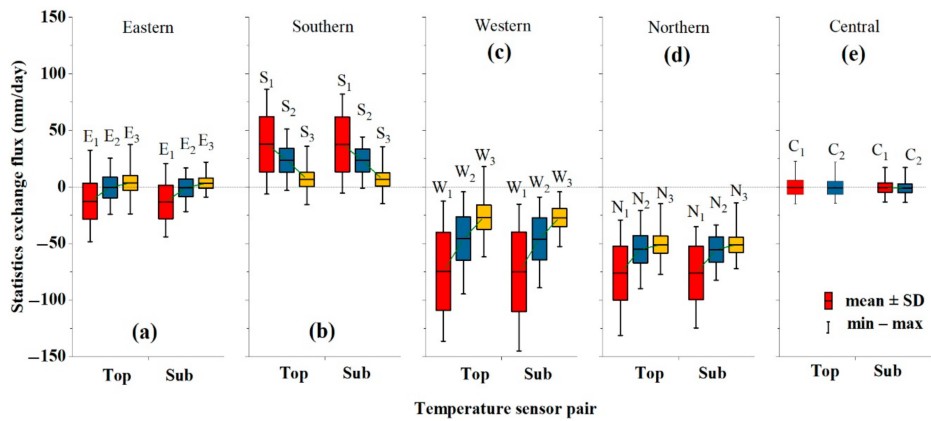

**Figure 8.** Statistical hyporheic fluxes at 14 sampling sites during 2016 using the VFLUX 2 model based on top (depths of 0 and 5 cm) and sub (depths of 5 and 10 cm) sensor pairs. Positive flux values represent downward flow and negative values represent upward flow. Site locations see Figure 1c.

In comparison, the estimated hyporheic fluxes based on the sub-pair matched well with those based on the top-pair, but their variations were relatively small (Figure 7). Overall, averages and variations of the hyporheic flux declined with offshore distance and reached almost zero at central sites. Furthermore, most of daily hyporheic flux variations from top-pair, which generally varied wider and frequently at most sites (i.e., $E_1$, $E_2$, $E_3$, and $W_3$), were not fully synchronized with those from sub-pair although their seasonal distributions matched well.

## 4. Discussion

### 4.1. Comparative Analysis of Estimated Hyporheic Fluxes Based On the Two Sensor Pairs

The estimated hyporheic fluxes based on the sub-pair were slightly different from those based on the top-pair, with lower daily and yearly fluxes and narrower variations (Figures 7 and 8), possibly resulting from thermal cycles in the shallow sediments by local water cycles. Local water cycles always occur in shallow sediments because of wave, wind current, and change of pressure head difference between lake water and groundwater [24,44]. Smith et al. [45] also reported that water moves back and forth between the surface and subsurface during hyporheic exchange, a process that influences the discharge quantity. In this case, the top-pair was possibly subjected to local water cycles in the shallow sediments and, consequently, to more frequent heat transferring from bottom water. Qin et al. [34] have reported that the disturbance depth in sediments caused by wind currents could reach 5–10 cm in Lake Taihu.

Overall, it can be considered that the estimated hyporheic fluxes based on the sub-pair could represent the long-term exchange flux between shallow groundwater and lake, whereas those based on the top-pair could represent the short-term and iterative water exchange conditions across the SWI.

Many studies have reported that the grid distance ($\Delta z$) between temperature sensors should be considered for improving the exchange flux estimation [4,19]. In general, those in upwelling and/or low flux conditions need to set a small $\Delta z$ (e.g., 5.0 and 2.5 cm and even smaller) pair sensor to effectively record the temperature oscillations, whereas those in downwelling and/or high flux conditions need to set a relatively large $\Delta z$. Gordon et al. [43] reported that fewer than 35% of the flux calculations were within the model sensitivity range at the shallowest center-of-pair depth in a strong water flux and downward direction with a $\Delta z$ of 5 cm, where vertical flux was high. Compared to some rivers or streams, either the upwelling or downwelling fluxes in the hyporheic zone of Lake Taihu are relatively low ($-136.5$ to $86.4$ mm d$^{-1}$). An additional running (not shown here) of VFLUX 2 model also show the estimated fluxes based on the sensor pair at 0 and 10 cm depths ($\Delta z = 10$ cm) matched well with those based on other two $\Delta z = 5$ cm sensor pairs. Sensitivity analysis by VFLUX 2 model also show that no time-steps had an $A_r \geq 1$ and almost 100% of flux values were calculated within the model sensitivity range. Thus, a $\Delta z$ of 5 cm was completely feasible to catch the apparent $A_r$s (e.g., averages of 0.18 at $E_1$, 0.27 at $S_1$, 0.11 at $W_1$, and 0.16 at $C_1$, respectively) and the estimated fluxes in this study were accredited either in the downwelling or upwelling conditions.

### 4.2. Sensitivity Analysis of Thermal Dipersivity on the Estimated Fluxes

Solid, liquid, and gaseous materials in soil/sediment are common carriers of heat as well as organic matter, in particular flowing mediums like water and vapor. However, the effect of physical features of the sediment, discussed in many published studies, will not be discussed here. Furthermore, the physical features, heat capacities, and thermal conductivities of the saturated sediment spatially varied little both horizontally and vertically in the hyporheic zone of Lake Taihu (Figure 3). No significant difference in thermal features as well as physical features was found between sites and depths ($p < 0.05$). In this case, the physical features of the sediment in the lake should not be considered as a dominant factor influencing the temperature–depth profiles in the hyporheic zone. In addition, air temperature at sampling sites exhibited slight difference (within about 2 °C),

leading to only very small differences of temperature at the interface (Figures 4 and 5) between sampling sites.

However, the hyporheic flow has a critical effect on heat transfer depth and temperature differences between the interface and a deeper depth [18]. Changes in amplitude with depth are most sensitive to seepage rates although under low-flow conditions. In general, temporal temperature at the SWI varied wider and frequent, caused by large seasonal or daily variation of air temperature. Downwelling hyporheic flow transferred much more heat into the hyporheic zone and resulted in wide variation ranges of temperature at greater depths (e.g., at southern sites, Figure 5d–f). Inversely, the temperature at a deeper depth (e.g., 5 or 10 cm) varied within a relatively narrow range at western and northern sites (Figure 5g–l) due to upward hyporheic flow with relatively stable temperature in shallow groundwater. Furthermore, as the hyporheic flux increased, the hyporheic flow transferred more heat and its pulse into a deeper depth in a downwelling zone or retarded at a shallower depth in an upwelling zone. Compared to some rivers or streams, relatively lower fluxes ($-136.5$ to $86.4$ mm d$^{-1}$) in the hyporheic zone of Lake Taihu significantly decreased the thermal pulse penetrating from surface (Figures 5 and 6). Kim et al. [46] have reported that it is reasonable for a hyporheic flux (depending on the hyporheic zone depth) which is decreasing with increasing downwards or upwards flow paths.

Sensitivity analysis (Table 1) shows that the thermal dispersivity $\beta$ had a large effect on the thermal diffusivity $\kappa_e$ and then the estimated fluxes but non-change in orders of flux magnitude, although under the relative low value and narrow range of flux conditions. VFLUX 2 model shows that inversely estimated parameter $\kappa_e$s based on the Mccallum and Luce methods [47,48] were slightly higher than those measured values under steady flow conditions due to transient flow effects, and temporally varied. However, no apparent changes (<10%) of estimated flux occurred when those estimated $\kappa_e$ were used to estimating the fluxes again, indicating a relatively reasonable setting of the initial $\beta$ value in this study. Gordon et al. [43] also reported a sensitivity analysis of $\beta$ (0 and 0.1 m) with median flux values of $1.02 \times 10^{-5}$ m·s$^{-1}$ and $1.07 \times 10^{-5}$ m·s$^{-1}$, respectively. However, it is apparent that the effect of $\beta$ increased with the flux magnitude and increased larger in an upwelling condition (e.g., at $W_1$). This particularly amplified the peak-valley values of estimated flux (e.g., during winter and spring at $W_1$ or during summer and autumn at $S_1$) and changed the seasonal temporal distribution of estimated flux. This further explained the large effect of hyporheic flow and the thermal dispersivity $\beta$ should be considered seriously although under relatively low flux conditions. Other in situ methods such as pressure difference method should be used to further verify the exchange flux and corresponding parameters.

**Table 1.** Sensitivity analysis for thermal dispersivity ($\beta$, m) on the estimated flux (mm d$^{-1}$) ranges and mean values (in parentheses) at sites $E_1$ (potentially alternative downwelling and upwelling), $S_1$ (downwelling), $W_1$ (upwelling), and $C_1$ (almost static flow) of Lake Taihu.

| Site | Sensor Pair | Base Value $\beta$ = 0.001 | Low Value $\beta$ = 0.0 | High Value $\beta$ = 0.01 |
|------|-------------|----------------------------|--------------------------|----------------------------|
| $E_1$ | 0 and 5 cm | $-48.7 \sim 32.4$ ($-12.8$) | $-46.6 \sim 31.5$ ($-12.3$) | $-82.1 \sim 42.6$ ($-20.3$) |
|       | 5 and 10 cm | $-44.1 \sim 20.6$ ($-13.3$) | $-42.2 \sim 20.0$ ($-12.8$) | $-73.2 \sim 27.8$ ($-20.9$) |
| $S_1$ | 0 and 5 cm | $-6.2 \sim 86.4$ (37.7) | $-5.9 \sim 84.6$ (36.8) | $-9.4 \sim 106.1$ (49.5) |
|       | 5 and 10 cm | $-5.6 \sim 82.0$ (37.7) | $-5.4 \sim 80.3$ (36.7) | $-8.6 \sim 101.3$ (49.4) |
| $W_1$ | 0 and 5 cm | $-136.5 \sim -12.3$ ($-74.5$) | $-128.9 \sim -11.9$ ($-70.9$) | $-288.2 \sim -18.5$ ($-137.4$) |
|       | 5 and 10 cm | $-145.0 \sim -15.3$ ($-75.0$) | $-137.1 \sim -14.7$ ($-71.4$) | $-301.0 \sim -22.5$ ($-138.0$) |
| $C_1$ | 0 and 5 cm | $-13.2 \sim 17.4$ ($-0.6$) | $-12.7 \sim 16.9$ ($-0.6$) | $-20.1 \sim 29.5$ ($-1.0$) |
|       | 5 and 10 cm | $-15.1 \sim 22.5$ ($-0.3$) | $-14.4 \sim 21.8$ ($-0.3$) | $-24.7 \sim 32.4$ ($-0.6$) |

*4.3. Hyporheic Flow Pattern Analysis*

Although only 3 nearshore sites were selected to study at each zone of Lake Taihu, the estimated hyporheic flow direction and flux magnitude in this study still represent well an overall situation about the interaction between the lake and shallow groundwater. Overall terrain in the watershed determines the overall flow direction from northwest to southeast (Figure 1b). The rainfall in the basin is usually rich (averagely 1181 mm per

year) particularly during summer season, generally remaining a shallow groundwater table below land surface. The hyporheic flow patterns in the nearshore hyporheic zone as described above agree with the overall flow direction in the basin and input/output locations at Lake Taihu, which are subject to the north Yangtze River and western and southwestern hilly topography (Figure 1b). Due to relatively higher groundwater level at western and northern sides of Lake Taihu [49], upwelling hyporheic fluxes occurred at northern and western sites were reasonable. At the eastern side of Lake Taihu, most groundwater level approaches to the lake level and varies seasonally due to precipitation and water level control in other water bodies of this region. This leads to an alternating upwelling and downwelling hyporheic flow at the eastern sites. These flow patterns in the nearshore hyporheic zone also agree with those reported earlier by Li et al. [40,50] on the basis of nutrient–and chloride–depth profiles.

Relatively high discharge fluxes at western and northern sites appeared mainly during winter and spring seasons, and relatively higher downwelling fluxes were appeared at southern sites. In comparison, groundwater discharge had a relatively higher contribution to Lake Taihu during winter and spring. Naranjo et al. [24] reported from a large oligotrophic sub-alpine lake, Lake Tahoe (USA), that groundwater discharge are temporally variable due to seasonal changes in recharge within the watershed, wave action, and lake stage, and groundwater discharge was enhanced by the seasonally-low lake stage and episodic recharge caused by precipitation falling as rain in the watershed. The temporal changed hyporheic flow patterns verified in this study also agreed well with other studies [28,29,40], reporting that the sediment of Lake Taihu spatial-temporally served as a source or sink of nutrients.

In addition, most groundwater–surface water exchanges concentrated in the nearshore hyporheic zone (Figure 7), where a relatively higher conductive medium and pressure head differences existed [17,51]. Shaw and Prepas [51] reported that seepage fluxes to a lake decreased exponentially with offshore distance. At the central zone of Lake Taihu (i.e., $C_1$ and $C_2$), no apparent upwelling/downwelling hyporheic flow exhibited at the deep depth except the surface temporal local water cycle possibly driven by external force disturbances (e.g., wind currents) [29,34,44]. However, results in this study mainly focused on the vertical part of the exchange flow, only representing a part of the water interaction between lake and groundwater. An inclination of lake bed usually exists at nearshore zone, which generally results in a large division of exchange flux in a non-vertical direction [44]. For a further study to obtain a better evaluation of the exchange flux, pair sensors should be considered to set along with the potential flow path and/or a vertical two-dimensional model needs to be used in the future.

### 4.4. Management Implications

In the past, the nutrients through hyporheic flow had scarcely been considered in Lake Taihu, whereas those directly released from surface sediments by disturbances (e.g., wind currents) were frequently reported and were admitted an important pollution source for Lake Taihu [29,30,32,34,41]. However, the nutrients transported by hyporheic flow also chronically contributes much to the lake particularly the nearshore zone serving as an active sink/source. For example, at the western and northern regions of Lake Taihu, the intensified agricultural and developed cities resulted in more pollution leaching into soil and shallow groundwater which closely connect the nearshore lake water by the hyporheic zone. Li et al. [40] previously reported that the concentrations of total nitrogen in the hyporheic zone (0–100 cm) ranged from 1.40 to 4.26 mg/L (mean 2.62 mg/L) at western sites and from 1.20 to 3.75 mg/L (mean 2.44 mg/L) at the northern sites, and concentrations of total phosphorus ranged from 0.28 to 1.24 mg/L (mean 0.54 mg/L) at western sites and from 0.23 to 1.17 mg/L (mean 0.61 mg/L) at the northern sites, respectively. These nutrients continuously enter the lake through groundwater and may seasonally contribute to additional sources of nutrients to the nearshore water. Additionally, many artificial wetlands for intercepting nutrients originating from agricultural lands and fishponds along

lakeshore should be drawn attention because these nutrients retaining in these waters would also slowly and continuously transport into the lake through the hyporheic zone.

It is necessary to collect more data to determine the source of nutrients in shallow groundwater to determine what management activities to prioritize. Some important management considerations, including discrimination of groundwater discharge and local water cycles and their associated nutrients, should be paid more attention in water quantity and quality estimating and modeling in the future.

## 5. Conclusions

Hyporheic flow pattern, including direction and flow velocity, was a major influencing factor of thermal transfer in the hyporheic zone. Observed temporal temperature distributions throughout 2016 differed between sampling sites and with zones in the Lake Taihu (Eastern China) due to diverse hyporheic flow patterns. Hyporheic fluxes estimated using VFLUX 2 model, based on the observed temporal temperature amplitudes, showed that downwelling flow occurred in the southern zone of Lake Taihu, upwelling flow occurred in the western and northern zones, and alternating-direction flow occurred in the eastern zone, but no apparent hyporheic flow existed in the central zone. Upwelling hyporheic fluxes at western and northern zones exhibited relatively high values during winter and spring seasons. Despite relatively small hyporheic fluxes, attention should still be paid to their concomitant nutrients, particularly on the western and northern nearshore water quality of Lake Taihu. These identified spatiotemporal exchange flow patterns and fluxes would further enhance the consideration of internal pollution in water quality modeling and improve lake management.

The temperature amplitude ratio method can be used to estimate the exchange flux and was suitable for low flux conditions (either upwelling or downwelling) when the distance between upper and lower sensors was 5 cm. However, the thermal dispersivity, which is more sensitivity for upwelling flow conditions, should still be considered seriously although under low flux conditions. A better evaluation of the exchange flux near lakeshore between surface and groundwater needs optimized installing of temperature sensors according to the potential flow path and/or a vertical two-dimensional flow and heat transport model in the future.

**Author Contributions:** Conceptualization, Y.L.; methodology, Y.L. and N.L.; software, N.L. and J.F.; validation, N.L.; formal analysis, N.L.; investigation, J.F., J.Q. and Y.S.; resources, Y.L.; data curation, N.L., J.F., J.Q. and Y.S.; writing—original draft preparation, N.L.; writing—review and editing, Y.L.; visualization, J.F., J.Q. and Y.S.; supervision, Y.L. All authors have read and agreed to the published version of the manuscript.

**Funding:** This research was funded by the National Natural Science Foundation of China (grant numbers: 51879081 and 51579074), the National Key Research & Development Program of China (grant number 2018YFC0407906), and the Priority Academic Program Development of Jiangsu Higher Education Institutions (PAPD).

**Institutional Review Board Statement:** Not applicable.

**Informed Consent Statement:** Not applicable.

**Data Availability Statement:** The data presented in this study belong to NSFC and presently are not publicly available due to their sensitivity.

**Conflicts of Interest:** The authors declare no conflict of interest.

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
