# Peer review of "Temporal Temperature Distribution in Shallow Sediments of a Large Shallow Lake and Estimated Hyporheic Flux Using VFLUX 2 Model"

_water, doi:10.3390/w13030300_

Round 1

Reviewer 1 Report

I think that the name amplitude for the deviations or fluctuations of temperature is not adequate. But, I am a fluid mechanic expert, and fluctuations I think is more adequate. I see 25 degrees amplitude in temperature and fluctuations of about 4 degrees. Fluctuations can come from physical behavior but also for instrumental errors. Amplitude of the wave of temperature come from changes of climate temperature. 

This is only a comment!

Reviewer 2 Report

In my  opinion  the manuscript  has a general good level  of quality and  it can be accepted  for publication. 

Author Response

Point 1: In my opinion the manuscript has a general good level of quality and it can be accepted for publication.

Response 1: We are very grateful to the referee for taking the time to review our manuscript. We have made some revisions for further improving the manuscript. Details about revisions are shown in the returned manuscript.

Reviewer 3 Report

Discuss the accuracy of equation 2 in the estimation of the diffusivity coefficient.

Is Cw in equation 1 different from Cw in equation 2? If not, why is it defined twice? provide the value in the first definition

Provide value for beta (thermal dispersivity)

Discuss the effect of turbulence on beta

Provide value for lambda0 (thermal conductivity of the bulk saturated medium)

Provide value C (Volumetric heat capacity of saturated sediment)

Improve the introduction by providing a more extensive background

Discuss the gap in the literature and how this paper addresses the gap

Explain how the heat transport equation has been solved numerically (equation 1).

Discuss how grid sensitivity analysis is performed in solving the heat transport equation (equation 1)

Discuss in more detail how the sampling locations were selects (the logic and methodology) and the impact of those choices on the generality of the results

Discuss in more detail how the results can be extended to other locations and how general are the results

Discuss the strengths and limitations of the employed approach in more details
Explain Hatch amplitude

Provide more background on VFLUX 2, how does it estimate the flux, how accurate it is

Reviewer 4 Report

Congratulations for the well-performed research and its presentation. It was a pleasure to read the manuscript, which also is about a highly interesting topic. Moreover, in the broader context it concerns the crucial environmental problem of deteriorated lake water quality that has to be addressed. 

I found only few minor issues:

  1. page 11, line 299: 'sub-pair pair', I think one pair is sufficient here
  2. page 12, lines 341-343: the authors argue here with higher terrain and surface altitude; however the decisive variable here is the groundwater level or piezometric head. The groundwater level could be aligned with topography, but due to the various activities in the lake vicinity (artificial wetlands, infiltration, withdrawal) that easy relation may likely be disturbed. I suggest to exchange to change the reasoning in these two sentences to address the groundwater and not the topography
  3. page 12, line 368: 'schlepped', I suggest to use the more common term 'transported'

Round 2

Reviewer 3 Report

The authors have addressed my comments.